# Anabolic Bone Stimulus Requires a Pre-Exercise Meal and 45-Minute Walking Impulse of Suprathreshold Speed-Enhanced Momentum to Prevent or Mitigate Postmenopausal Osteoporosis within Circadian Constraints

**DOI:** 10.3390/nu13113727

**Published:** 2021-10-22

**Authors:** Qingyun Zheng, Thomas Kernozek, Adam Daoud-Gray, Katarina T. Borer

**Affiliations:** 1School of Kinesiology, The University of Michigan, Ann Arbor, MI 48109, USA; adam.i.daoud@gmail.com (A.D.-G.); katarina@umich.edu (K.T.B.); 2School of Physical Education, Henan University, Kaifeng 475004, China; 3Physical Therapy Program, Department of Health Professions, University of Wisconsin-La Crosse, La Crosse, WI 54601, USA; tkernozek@uwlax.edu

**Keywords:** postmenopausal osteoporosis, exercise, nutrition, CTX, CICP, osteocalcin, bone-specific alkaline phosphatase, PTH, momentum, impulse

## Abstract

Osteoporosis currently afflicts 8 million postmenopausal women in the US, increasing the risk of bone fractures and morbidity, and reducing overall quality of life. We sought to define moderate exercise protocols that can prevent postmenopausal osteoporosis. Our previous findings singled out higher walking speed and pre-exercise meals as necessary for suppression of bone resorption and increasing of markers of bone formation. Since both studies were amenable to alternate biomechanical, nutritional, and circadian interpretations, we sought to determine the relative importance of higher speed, momentum, speed-enhanced load, duration of impulse, and meal timing on osteogenic response. We hypothesized that: (1) 20 min of exercise one hour after eating is sufficient to suppress bone resorption as much as a 40-min impulse and that two 20 min exercise bouts separated by 7 h would double the anabolic effect; (2) early morning exercise performed after eating will be as effective as mid-day exercise for anabolic outcome; and (3) the 08:00 h 40-min. exercise uphill would be as osteogenic as the 40-min exercise downhill. Healthy postmenopausal women, 8 each, were assigned to a no-exercise condition (SED) or to 40- or 20-min exercise bouts, spaced 7 h apart, for walking uphill (40 Up and 20 Up) or downhill (40 Down and 20 Down) to produce differences in biomechanical variables. Exercise was initiated at 08:00 h one hour after eating in 40-min groups, and also 7 h later, two hours after the midday meal, in 20-min groups. Measurements were made of CICP (c-terminal peptide of type I collagen), osteocalcin (OC), and bone-specific alkaline phosphatase (BALP), markers of bone formation, and of the bone resorptive marker CTX (c-terminal telopeptide of type 1 collagen). The osteogenic ratios CICP/CTX, OC/CTX, and BALP/CTX were calculated. Only the 40-min downhill exercise of suprathreshold speed-enhanced momentum, increased the three osteogenic ratios, demonstrating the necessity of a 40-min, and inadequacy of a 20-min, exercise impulse. The failure of anabolic outcome in 40-min uphill exercise was attributed to a sustained elevation of PTH concentration, as its high morning elevation enhances the CTX circadian rhythm. We conclude that postmenopausal osteoporosis can be prevented or mitigated in sedentary women by 45 min of morning exercise of suprathreshold speed-enhanced increased momentum performed shortly after a meal while walking on level ground, or by 40-min downhill, but not 40-min uphill, exercise to avoid circadian PTH oversecretion. The principal stimulus for the anabolic effect is exercise, but the prerequisite for a pre-exercise meal demonstrates the requirement for nutrient facilitation.

## 1. Introduction

A decline in estradiol following the onset of menopause is accompanied by loss of bone mineral content (BMC) and density (BMD) and increased risk of osteopenia, osteoporosis, and bone fractures [1] which diminish the quality of life even when the fractures heal [2]. Between now and 2040, postmenopausal osteoporotic fractures in U.S. are projected to rise from 1.9 to 3.2 million, and associated medical costs from $57 to about $95 billion [3]. The prevailing approach to prevention of osteoporosis and bone fractures has entailed administration of estradiol, parathyroid hormone (PTH), and calcitonin, anti-resorptive drugs such as bisphosphonates and selective estrogen receptor modulators, and more recently, injections of anti-resorptive or anti-catabolic antibodies. Antibodies against bone cytokine RANKL (in drug Denosumab or Prolia) block RANKL stimulation of bone resorption, while antibodies against osteocyte product sclerostin (in drug Romosozumab or Evenity) block its inhibition of the bone anabolic pathway [4,5]. While they improve osteoporotic BMD and reduce fractures, all these approaches have long-term adverse side effects with use. Hormone replacement therapy with estrogen and progesterone and intermittent injections of PTH drug teriparatide are associated respectively with occasional development of breast [6] and bone cancer [7]. Anti-resorptive drugs can lead to osteonecrosis of the jaw and atypical fractures of the femur [8]. Discontinuation of the Denosumab injections produces a rebound effect with increased risk of multiple spontaneous vertebral fractures [9]. Discontinuation of Romosozumab reverses BMD gained during 12 months of treatment [10].

Exercise is an alternative approach to prevention of osteoporosis that facilitates healthy bone remodeling, a dynamic process that couples osteoclastic bone resorption with osteocytic and osteoblastic bone mineral accretion to repair microfractures as it replaces old bone with new [11]. Unfortunately, exercise generally, and particularly in postmenopausal women and older adults, produces at best very modest (1–3%) increases in bone mineral accrual [12,13] unless older athletes and adults engage in exceptionally high volumes of exercise with high ground- and joint-reaction forces [14,15,16,17].

We have attempted to examine moderate exercise paradigms suitable for sedentary non-athletic individuals that could prevent or mitigate postmenopausal osteoporosis. In the first study [18], we determined that 45 min of early morning exercise in healthy postmenopausal women preserved and slightly increased whole body and leg BMD after four months of walking on level ground, while slower walkers covering the same 4.8 km distance lost bone mineral. Our second study [19] demonstrated the essential role of nutrition in amplifying the osteogenic effects of exercise. Here, the type-2 diabetic postmenopausal women exercised for 40 min either at 08:00 and at 15:00 h before eating one of two daily meals or at 11:00 and 18:00 h, one hour after a meal. Increased concentrations of bone-formation marker CICP (C-terminal propeptide of type 1 collagen) were observed after both post-meal exercise bouts, but not when the same exercise was performed one hour before eating. A similar close link between pre-exercise meal and a bone anabolic response was recently confirmed in an animal study [20] where, in young mice, formation of new cortical bone in the mechanically loaded limb was facilitated by refeeding after a 16-h fast. Armed with the understanding that both a higher speed-enhanced momentum, and a pre-exercise meal are required for an osteogenic response, the aim of the present study was to fill the knowledge gap regarding the necessary duration of the exercise impulse and the potential circadian influence on the osteogenic outcomes. The interest in the necessary duration of osteogenic exercise reflected the contrast between the animal studies where a bone anabolic response could be achieved with a mechanical stimulus of very short duration compared to the 40- and 45-min exercise duration stimulating bone anabolism in our two previous studies. Forty daily mechanical load cycles produced an anabolic response in rat tibias and failed to increase with additional loading up to 380 cycles [21]. Only 5 to 10 jumps per day significantly increased the mass of rat femoral and tibial cortical area, and up to 100 more jumps per day were ineffective [22]. By contrast, the effective bone anabolic impulse in response to 40 min of exercise in our previous study involved between 4100 and 5200 steps [19], and we sought to find out whether exercise of half this duration would be as effective. In addition, we questioned whether two bouts of 20 min exercise would amplify the effect of a single consolidated exercise bout as a result of transient 6- to 8-h period of reduced bone mechanosensitivity during which added mechanical stimulation of animal bones is ineffective. When, 6 to 8 h later, the mechanosensitivity is restored, two exercise bouts spaced by that amount of time, double the magnitude of the anabolic effect [23,24]. We confirmed this phenomenon in the previous study [19], where each of the two 40-min exercise bouts spaced by 7 h and performed an hour after eating, increased the concentration of CICP, with the total increase exceeding the concentration increase of a single bout.

We hypothesized that:
(1)A 20-min exercise impulse of suprathreshold walking-speed-enhanced momentum, performed an hour after a meal, will be as anabolic as a 40-min exercise impulse; and two 20-min exercise bouts spaced 7 h apart would double the anabolic response to a single one;(2)Anabolic outcome of early morning (08:00 h) exercise performed an hour after eating will be as effective as a mid-day (15:00 h) post-meal exercise;(3)40-min exercise at 08:00 h and 20-min exercise at 08:00 and 15:00 h will be equally osteogenic whether performed uphill or downhill.

Out of concern for a possible circadian effect of different exercise timing in the present relative to the preceding study [19], we included timed measurements of anabolic hormones insulin, leptin, PTH, and growth hormone and of the catabolic hormone cortisol, because secretory responses of all of these hormones are affected by exercise [25,26,27,28]. The first four support bone formation, and cortisol can block it [29,30,31,32,33]. 

## 2. Materials and Methods

### 2.1. Subjects

Forty postmenopausal women were recruited from the University of Michigan clinical studies web page (UMClinicalStudies.org) and newspaper advertisements. Inclusion criteria were 50 to 65 years old, surgical or natural menopause with no menstrual periods for at least one year, no metabolic disease, body mass index (BMI) of 24 to 30 kg/m^2^, no hormone replacement therapy, non-smoker, absence of musculo-skeletal disabilities that would prevent treadmill walking, and sedentary status (<60 min of regular exercise per week). The study was conducted in accordance with the Declaration of Helsinki, and the protocols (HUM 00032227 on 8 October 2009, and HUM 00032700 on 22 January 2010) and informed consent forms were approved by the University of Michigan Medical School Institutional Review Board (IRB-MED).These protocols were registered as a clinical trial NCT04063813 with ClinicalTrials.gov after the start of subject recruitment in 2009 and its completion in 2012.

### 2.2. General Experimental Protocol

Subjects underwent preliminary health and fitness screens at the Michigan Clinical Research Unit (MCRU). The health screen included a fasting blood draw, health history, height and weight, body fat, and bone mineral parameters measurements by a dual-energy X-ray absorptiometry (DXA) scanning. Baseline fasting glucose served to confirm non-diabetic status, and baseline parathyroid hormone (PTH) and thyroid stimulating hormone (TSH) determinations were employed to assess possible hyperparathyroidism and hypothyroidism, respectively. A preliminary fitness screening assessed individual maximal aerobic effort. It consisted of a treadmill test at 4.8 km per hour with 2% slope increments every 3 min. To obtain oxygen consumption (ṼO_2_) and carbon dioxide production (ṼCO_2_), the subject was breathing through a mouthpiece and tubing into a gas meter using a Max II metabolic cart (AEI Technologies, Inc., Bastrop, TX, USA). Gas-meter calibration was done with pre-calibrated gas tanks. The criterion for maximal effort used was a respiratory quotient (ratio of ṼCO_2_/ṼO_2_) of 1. After matching by age, body weight, BMI, and aerobic fitness, subjects were stratified to five trials. A sedentary trial involved no exercise (SED), two downhill trials were performed at a −6° treadmill decline, involving either a single 40-min (40Down), or two 20-min (20Down) exercise bouts spaced 7 h apart. Corresponding uphill trials, a single 40-Up and two 20-Up were performed at an average +8° treadmill incline.

### 2.3. Study Design

A week after the fitness test, subjects were admitted to the MCRU at 06:30 h for a 16-h trial. At 06:45 h, an intravenous catheter, kept patent with sodium heparin, was inserted into an antecubital vein. At 07:00, subjects received 1 g calcium and a 600 IU vitamin D supplement with one oz of orange juice. Three weight-maintenance meals were provided at 07:00, 13:00, and 19:00 h. The exercise bouts were performed from 08:00 to 08:40 h in two groups (40 Down and 40 Up) and from 08:00 to 08:20 h in the morning and 15:00 to 15:20 h in the afternoon in two other groups (20 Down and 20 Up) (Figure 1).

During exercise, women wore dynamic in-shoe mechanosensitive insoles containing sensors that provided information on peak pressures to obtain the ground reaction force (GRF) and number of steps during walking. They also wore heart-rate-monitor chest bands and wrist watches for measurement of heart rates (Polar Electro, Bethpage, NY, USA). Blood (3 mL) was collected between 08:00 and 22:00 h at the following intervals: 8:00, 8:20, 8:40, 9:00, 10:00, 11:00, 13:00, 15:00, 15:20, 15:40, 16:00, 17:00, 18:00, 20:00, and 22:00 h.

### 2.4. GRF Manipulation and Measurements

Uphill treadmill slope elevation was used to reduce GRFs, and downhill treadmill slope to increase them relative to level-surface walking [34]. Downhill slope adjustment was accomplished through a treadmill modification, which entailed construction of a lever arm powered by a mechanical jack that raised the rear end of the treadmill to create a −6° angle treadmill slope [19]. Uphill exercise utilized treadmill elevation features. GRF measurements normal to the plantar surface during walking were recorded during each stepping cycle bilaterally at 50 Hz using the in-shoe pressure-measuring insoles (Novel Pedar, Novel Electronics, St. Paul, MN, USA) and the associated computer software (Pedar Professional, Novel Electronics, St. Paul, MN, USA). The pressure-measuring system was carried on a belt around the patient’s waist. Prior to walking, each sensor insole was zeroed using the Pedar software. Measurements taken during the last 6 min of the 40-min exercise were used to determine peak pressures in kilo pascals (KPa). Pressures from 99 sensors per insole were converted to total GRF in Newtons within the Novel Pedar software by multiplying each sensor value by the sensor size.

### 2.5. Exercise Intensity

Desired relative exercise intensity as percent of maximal effort was estimated during the first 10 min of exercise from the preliminary fitness tests and then standardized by adjusting the treadmill slope and walking speed using respirometry. Intensity-associated psychological stress during exercise was measured with Borg’s ratings of perceived exertion (RPE) scale [35]. The scale ranges from 6 to 20, and seven intensities are identified as, 7 = very, very light; 9 = light; 11 = fairly light; 13 = somewhat hard; 15 = hard; 17 = very hard; and 19 = very, very hard. RPEs, along with heart rates, were measured both during the preliminary aerobic fitness test and during exercise trials at 5-min intervals.

### 2.6. Meals

Three weight-maintenance meals were provided at 07:00, 13:00, and 19:00 h The breakfast was provided one hour, and the mid-day meal two hours, before exercise. Macronutrient composition was 60% carbohydrate, 15% protein, and 25% fat and provided 25%, 35%, and 40% of daily calories in the morning, midday, and evening meals, respectively. Foods to meet this composition were selected by MCRU dietitians (Table 1).

### 2.7. DXA Measurements

Areal BMD (g/cm^2^), BMC, and body composition were determined with a DXA apparatus (model Prodigy, Lunar Radiation Corporation, Madison, WI, USA) using the pencil beam mode. Scans included lumbar spine (L1 through L4) and hip (femoral neck, trochanter, Ward’s triangle, and femoral shaft). Coefficients of variation (CVs) for BMD measurements of the separate regions ranged between 1.5% (spine) and 2.0% (hip). The quality control program included weekly calibration studies.

### 2.8. Blood Collection

For determination of bone markers, hormones, and calcium, three-milliliter blood samples were collected into serum-separation tubes containing spray-coated silica and polymer gel (BD Vacutainer venous serum separation tubes, Hemogard, Fisher Scientific, Pittsburg, PA, USA). Serum was separated 30 min later by centrifugation at 2000 g. It was then stored at −80 °C for later hormone and bone marker determinations.

### 2.9. Markers of Bone Formation and Resorption

Bone formation markers measured were C-terminal propeptide of type I collagen (CICP), osteocalcin (OC) [36], and bone-specific alkaline phosphatase (BALP) [37]. The marker of bone resorption measured was C-terminal telopeptide of type-I collagen (CTX). Of particular interest was the ratio of markers of bone formation versus the marker of bone resorption as we expected it to reflect the overall osteogenic response. CICP was measured with enzymatic immunoassay and BALP with enzymatic immunometric assay (EIA) using kits provided by Quidel (Santa Clara, CA, USA). Metra-CICP enzyme immunoassay had a sensitivity of 0.2 ng/mL. Intra- and inter-assay CVs at three dose levels ranged between 5.5% and 7%. BALP EIA intra- and inter-assay CVs at three dose levels ranged between 3.9 and 7.6. OC was measured with Milliplex human bone panel 1A kit HBN1A-51K (Millipore Sigma, St. Charles, MO, USA). It utilized fluorescently labeled microsphere beads quantified in a Luminex 100 ^TM^ instrument. OC assay limit of detection was 39.3 pg/mL, and its intra- and inter-assay CVs were 5.6 and 10.7%, respectively. CTX was measured with serum CrossLaps enzyme immunological assay by Nordic Bioscience Diagnostics (also supplied by Quidel). Sensitivity of this assay was 20 pg/mL. Intra- and inter-assay CVs at three dose levels ranged between 5.0% and 8.1%. CICP, OC, and CTX were measured in all 40 study subjects, while BALP was measured in a subset of 25 subjects. 

### 2.10. Hormone and Calcium Measurements and Endocrine and Circadian Considerations

The basal concentrations of two hormones, TSH and PTH, were measured by the University of Michigan Chemistry Laboratory as part of qualification for inclusion in the study. The same laboratory measured total serum calcium concentrations to assess their action as a potential trigger of PTH concentrations. Intact PTH from the study samples was measured with solid-phase two-site chemiluminescent immunometric assay (DiaSorin, Vercelli, Italy). Intra- and inter-assay CVs for PTH at two dose levels were between 1.2% and 2.2% and 4.8% and 7.7%, respectively. Insulin and leptin were measured with RIA kits HL14-K and HL81HK, respectively (Linco Research, Millipore Corp, St Charles, MO, USA). The intra- and inter-assay CVs for insulin were 2.2% and 20.0%, and for leptin were 9.1% and 14.2%, respectively. GH was measured with a chemiluminescent Milliplex assay HPT-66K with a 4 pg/mL limit of detection (Millipore Corporation, St. Charles, MO, USA). Its intra- and inter-assay CVs were 5.4% and 14.9%, respectively. Cortisol was measured with a solid-phase radioimmunoassay (Siemens Medical Solutions Diagnostics, Los Angeles, CA, USA). Intra- and inter-assay CVs were between 3% and 5.1%, and 4% and 6.4%, respectively.

### 2.11. Statistical Analyses and Calculations of Mechanical Loading

Data are presented as means and SEMs. To eliminate inter-personal variability, bone markers, and hormones are presented as percent change from the 08:00 h concentration value. Hormone values were also log transformed to insure normal value distribution. Osteogenic bone marker effects were assessed both by recording the changes in their concentrations and by calculating osteogenic ratios for CICP/CTX, OC/CTX, and BALP/CTX. Number of walking steps was determined based on a user-written program from the peak total vertical ground reaction force of each Novel Pedar sensor insole when the force was greater than 50 N and when its loading duration was between 0.3 and 0.5 s [38]. Two relationships between force and time that do not have standard terms in physics, but appear to be relevant to data in this study, were calculated. The first one, momentum, relevant to interpretation of the results of the study which manipulated the velocity of walking [18], is derived from Newton’s second law of motion, where both the force (F = m*a) and momentum (Q) are products of body mass and its acceleration (with mass in kg units and acceleration in m/s). 

The second force–time relationship relevant in the present study for examining the importance of the duration of exercise bout is the impulse or force–time integral. Impulse is the area under the force–time curve for each step which depicts the sum of the effect of force over time (Figure 2) during the loading portion of the gait cycle. Impulse, measured in N*s, was determined using the trapezoid rule over the duration of the stance phase of walking [38] when the total ground reaction force from the sensor insole was greater than 50 N from the start to the end of each step. The individual samples of force were calculated from every third right footstep during stance and were multiplied by 0.02 s and then summed to calculate the impulse per step. Impulse measure is being applied in clinical studies where the health outcome is changed by a modification of either the force or duration of its application [39].

All statistical analyses were performed using Statistical Analysis Software (SAS version 9.4, SAS Institute, Cary, NC, USA). Between-group comparisons were performed as repeated-measures mixed-model ANOVA where the treatment and time effects and their interactions were analyzed as between-subject effects, and the values for each of 40 subjects, as within-subject random intercept. Where no overall between-group differences were seen, time slice effects indicated significant differences at specific time periods. As this analysis suggested a difference in the magnitude of bone marker and hormone changes between the two spaced postprandial periods and the two spaced exercise bouts, an analysis of the effects of timing of the first and second exercise bouts (EX1 and EX2) and of the first and second postprandial periods (PP1 and PP2) was performed on the respective areas under the curve (AUCs) using trapezoidal rule. The AUCs for the first exercise bout (EX1) was calculated for the time period between 08:00 and 11:00 h, and for EX2 between 15:00 and 18:00 h. The EX1 AUCs included the 2-hour exercise bout and one post-exercise hour to the onset of mid-day meal. EX2 AUC included the 2-h exercise bout and one post-exercise hour. The AUC for the first postprandial period (PP1) was calculated for the time period between 07:00 and 13:00 h between the morning and mid-day meal, and for PP2 between 13:00 and 19:00 h for the equivalent length of time after the mid-day meal. Overall between-group differences for the two exercise-period AUCs and for the two postprandial-period AUCs were calculated with mixed-model ANOVA using between-group EX1–EX2 and PP1–PP2 difference scores. Differences between any of the five groups were calculated with Tukey’s Studentized range t test that adjusts for multiple comparisons, and the differences between any exercising group and the sedentary group was established with Dunnett’s *t*-test that also adjusts for multiple comparisons. Within-group comparisons of differences between EX1 and EX2, and between PP1 and PP2, were calculated with the paired *t*-test procedure. AUC comparisons were also performed for the hormones in which measurements were sufficiently frequent for the timing-effect analysis. Our intent was to find out, where possible, whether the timing of exercise and postprandial periods produced hormonal changes that coincided with changes in the ratios of markers of bone formation and bone resorption. Figure graphics were produced with GraphPad Prism 8.4.2 software (GraphPad Software, Inc., San Diego, CA, USA).

## 3. Results

Subject characteristics are outlined in Table 2. None of the tested variables differed between the five groups, but the difference in the duration of menopause approached significance (*p* = 0.064). Mean fasting serum glucose was normal at 93.9 ± 2.9 mg/dl. 

### 3.1. Exercise Outcome

Both the peak GRFs, relative effort, and walking speeds manipulated through different treadmill slopes were significantly different between uphill and downhill walking (Table 3). The relative effort in uphill groups was 72% of maximal and significantly higher than the 64% effort in the two downhill groups. Walking speed, distance travelled, and relative effort all were between 43 and 55% higher in Down than in Up trials, and peak stepping pressures, both absolute and relative, and the number of steps taken in 40 min, were all between 20 and 30% higher in Down relative to Up groups. Peak GRFs of the downhill groups were overall 31% higher than GRFs of uphill groups, by 42.2% and 21.1% for 40-min and 20-min groups, respectively. Walking momentum of the two Down groups was between 37.1% (40-min trials) and 34.5% (20-min trials) higher than that of Up groups, while the 40-min impulse or force–time integral of both Down and Up groups was double that of 20-min Down and Up groups. RPE values of subjective stress were not different between Up and Down groups and ranged between 10 (light) and 12 (somewhat hard). Trial heart rates (HR) were variable but only 8% higher in Up than in Down groups (F_(df = 3,28)_ = 10.33; *p* < 0.0001).

### 3.2. Bone Marker Responses to Timing of Exercise and Meals 

Bone marker serum concentrations in response to exercise and meals revealed significant treatment, time, and interaction effects for CICP (Figure 3A) and treatment and time effects in CTX, (B), but no changes in OC (C) and BALP (D) (Table 4). Serum CICP concentration was significantly higher in 40 Down, than in SED group (*p* = 0.0185). CTX was significantly lower in 40Down group than in the 20 Up group (*p* = 0.0011). In both CICP and CTX, there were significant between-group differences at time slices immediately following the first and second exercise bouts (Table 4). 

### 3.3. Changes in the Ratios of Markers of Bone Formation and Marker of Bone Resorption to Timing of Exercise and Meals 

Treatment and time effects and their interaction were significant for CICP/CTX and OC/CTX (Figure 4A,B, respectively). For BALP/CTX, only the effect of time and its interaction with treatment were significant (Figure 4C) (Table 4). For CICP/CTX, 40 Down group ratio was significantly higher than SED (*p* = 0.0309) and 20 Up (*p* = 0.0108) trials (Figure 4A). For OC/CTX trial, 40 Down ratio was significantly higher than the SED (*p* = 0.0376), 20 Up (*p* = 0.0033), and 40 Up (*p* = 0.0392, Figure 4B) bouts. In all three bone marker ratios, there also were significant differences between groups at time slices after the second exercise bout (Figure 4A–C, Table 4).

### 3.4. Effects of Timing of EX1 and EX2 Exercise AUCs, and of PP1 and PP2 Postprandial AUCs on Markers of Bone Formation, the Marker of Bone Resorption, and Their Ratios

Since the ratio of markers of bone formation and the marker of bone resorption showed some osteogenic effect after the second, but not the first, daily 20-min exercise bout and meal (Figure 4), analyses were done on the effects of timing of the two daily exercise bouts and postprandial periods by comparing their AUC difference scores after Tukey and Dunnett adjustments for multiple comparisons. All three markers of bone formation revealed significant between-group differences in timing of exercise AUCs and only a trend for CTX (Table 5). In general, the 40-min Down EX1 exercise AUCs were most frequently significantly higher than EX2 AUCs compared to the other trial AUCs. The 40-min Down EX1 exercise AUCs were higher than SED AUCs in BALP (Figure 5B), CTX (panel C), CICP/CTX (panel D), and OC/CTX trials (panel E) (Table 5). They were also higher than 20 Up exercise AUCs for OC (panel A), OC/CTX (panel E), and BALP/CTX (panel F), and for 20 Down AUCs in OC/CTX trial (panel E). Only in CICP were exercise 20 Down AUCs greater than 20 Up AUCs (Table 5).

In within-group comparisons, the EX1 AUCs were significantly higher than the EX2 AUCs for OC, BALP, and CTX in three 40 D trials (Figure 5A–C, respectively). Because the reduction in the second exercise AUCs was greatest for CTX, all three marker ratios had second exercise AUCs greater than the first one (Figure 5D–F, respectively and Table 5). 

In postprandial AUC analyses, there was a single between-trial difference between the 40 Down OC/CTX and SED PP AUCs (Figure 6C and Table 6). In within-trial PP AUC comparisons, CTX PP1 AUCs were significantly higher than PP2 in all four exercise trials as well as in the SED trial (Figure 6A). As a result, PP2 AUCs were higher that PP1 AUCs for all three bone marker ratios (Table 5 and Figure 5B–D).

### 3.5. Hormone and Calcium Measurements 

Overall, there was no apparent pattern in serum GH, leptin, insulin, and cortisol concentrations that would suggest that they played a role in existing exercise and diet effects on bone markers and their ratios. A significant treatment effect was apparent only for PTH (Figure 7A), while the time and interaction effects were seen for all hormones but leptin (B, D–F), and a time effect only for leptin (panel C) (Table 7). Serum PTH, GH (Figure 7A,B, respectively), and cortisol (panel E) concentrations increased during the two bouts of exercise, at which times there also were significant time-slice effects. Serum leptin differences appeared during the late afternoon (panel C), insulin concentration increased at mealtimes (panel D), and calcium concentration declined in the exercising trials below that in the SED trial (panel F). 

Post-exercise pattern of PTH concentrations differed between the 40 min downhill and uphill groups (Figure 7A). Although both groups were characterized by an exercise-associated peak one hour post exercise, mean post-exercise PTH concentration was persistently higher after uphill than after downhill exercise (51.0 ± 2.5 vs. 43.4 ± 1.9 pg/mL respectively, t_(df = 14)_ = 6.198, *p* < 0.0001).

Overall exercise and postprandial effects were seen only in insulin (Table 8). Insulin also had the greatest number of significant between-group AUC differences, four in the exercise analyses (Figure 8B) and five in postprandial AUC analyses (Figure 8E), while PTH and cortisol had a single exercise AUC group difference (panels A and C) and no postprandial ones (Table 8). 

Notable differences in timing of exercise and postprandial AUCs between morning and afternoon were seen only in PTH, insulin, and cortisol (Table 8) of which the morning AUCs were higher than afternoon AUCs for PTH and insulin, while the reverse relationship was found in cortisol (Table 8).

## 4. Discussion

The purpose of this study was to define a moderate exercise paradigm suitable for sedentary non-athletic women that could prevent or mitigate postmenopausal osteoporosis. We had two preliminary cues defining the necessary exercise parameters for producing an anabolic response. The first one was the importance of walking speed. Bone mineral loss could be prevented with four months of faster walking which generated higher speed-enhanced momentum. In this study, healthy postmenopausal women were recruited and supervised while walking daily for 4 months an assigned distance of 4.8 km at a commercial mall starting at 06:30 or 08:00 h. Faster walking at 6.4 km/h or 1.8 m/s on level ground suppressed bone mineral loss and moderately increased BMD in legs and whole body, while slower walking at 5.5 km/h or 1.5 m/s failed to do so [18]. Fast walkers completed a 4.8 km course in 45 min, while slow walkers took 53 min. This suggested that the anabolic effect was probably caused by increased walking speed and the resulting speed-enhanced increase in momentum (137.5 ± 4.8 kg*m/s in fast walkers vs. 117.8 ± 4.5 kg*m/s, t_(df = 23)_ = 2.526, *p* = 0.019 in slow walkers), a 45-min impulse, and even the early morning exercise onset [18]. The second cue was that eating a meal one to two hours before exercise was necessary to increase the concentration of a bone-formation marker CICP [19]. In this study, 40-min of treadmill walking either uphill or downhill twice a day at 11:00 and 18:00 h, one or two hours after eating a meal, produced equivalent increases in CICP bone marker concentration in diabetic postmenopausal women. The same exercise performed at 08:00 and 13:00 h, but before eating a meal, produced no anabolic response. This then suggested that a pre-exercise meal was required for exercise to exert an anabolic response. However, we did not control for the possible confounding factor of a difference in the circadian timing of exercise. Each of our conclusions was open to alternative interpretations. Therefore, the first among unresolved issues regarding exercise-induced anabolic bone responses, was the necessary duration of exercise impulse. The effective 45.4 min impulse in fast-walking women on the mall was an accidental consequence of setting up an arbitrary walking distance of 4.8 km that required at least that amount of fast walking time. Since slower walking for as long as 53 min produced no response, it was clear that effectiveness of the exercise impulse depended on a walking speed in excess of 6 km/h. This did not clarify the importance of impulse duration, as we implemented a 40-min walking impulse in the present and the previous [19] study for consistency with the first [18] study. The present study was an attempt to resolve the impulse duration uncertainty. The relative duration of exercise impulse was tested in Hypothesis 1 along with the role of 7-h spacing of exercise bouts to potentially double the anabolic effect by exceeding the period of bone mechanosensitivity. The consistency of anabolic response to required post-meal exercise was tested in Hypothesis 2 to see whether it replicated the findings of the previous study [19]. A potential circadian interference with anabolic effects was tested in Hypothesis 3 by comparing the equivalence in the bone marker outcomes to 40-min and 20-min uphill and downhill exercise bouts initiated at 08:00 h to see whether they were as effective as in the previous study [19] where exercise was initiated at a different circadian time of 11:00 h.

### 4.1. Testing of Hypothesis 1: The Importance of Exercise Impulse for Osteogenic Response

The key result of Hypothesis 1 testing was our discovery of the relevance of the duration of exercise bout or impulse (Figure 2) in producing a bone anabolic response. As an added control for the possible importance of ground-reaction forces in bone anabolic response, all exercise bouts in the present and previous [19] study were carried out both uphill and downhill, as the slope of the surface for walking or running affects the magnitude of GRF loading [34]. Our results were unambiguous. In the present study, only the 40-min impulsive loading of downhill exercise increased the concentrations of CICP (Figure 3A) and CTX (panel B) and osteogenic ratios of CICP/CTX, OC/CTX, and BALP/CTX (Figure 4A–C). In the previous study [19], both the 40-min downhill and the 40-min uphill exercise bouts were anabolic. The anabolic CICP outcome [19] was not affected by a 25% difference in GRFs resulting from a difference in treadmill slopes. The suppression of bone mineral loss by fast walkers in the mall study was achieved with a 45.4-min impulsive loading on the level ground, but not with a longer 53-min impulsive loading at a slower speed [18]. In the present study, a 20 min impulse, either uphill or downhill, failed to increase the concentrations of CICP, CTX (Figure 3A,B, respectively), or the osteogenic ratios of CICP/CTX, OC/CTX, and BALP/CTX (Figure 4). The impulsive loading for suppressing CTX concentration by walking was 40min long and amounted to 2,662,800 ± 110,880 N*s in the present downhill study. To increase CICP concentration, the 40-min impulsive loading was 2,651,664 ± 144 N*s in the previous study [19], and it was 45 min or 2,327,018 N*s for fast walkers in the mall study [18]. These osteogenic impulses required between about 5209 ± 100 steps downhill and 4568 ± 82 steps uphill (Table 3). By contrast, 20-min impulsive loading of downhill (1,220,520 ± 58,440 N*s) or uphill walking (1,007,880 ± 36,960 N*s) were not sufficiently long to stimulate an anabolic bone response despite their generating between 2599 and 2052 downhill and uphill steps, respectively. Therefore, the prerequisites for an anabolic bone effect requires exercise speed of greater than 6 km/h, increased speed-assisted momentum of about 130 kgm*s, and an impulse of about 2,600,000 N*s lasting 45 m for exercise on level ground and 40 min for exercise at a −6° downhill slope. In clinical medicine, the impulse measures the effectiveness of a treatment with components of both intensity and duration each of which can affect the outcome [39]. We now provide evidence that a 40- to 45-min impulse is required for an osteogenic effect of exercise, while cutting it in half is ineffective.

A second component of Hypothesis 1 posited that 20-min exercise bouts separated by 7 h will double the anabolic response of a single 40-min bout by restoring the mechanosensitivity lost during the initial bone loading. We based this hypothesis on the animal studies where a very small number of mechanical stimuli were found to effectively stimulate osteogenesis followed by several hours of refractoriness. Separating a single effective mechanical stimulus in these studies into two components spaced by about 7 h, restored mechanosensitivity and produced twice the amount of bone growth as with the single consolidated mechanical stimulus [23,24]. We therefore assumed that our 40-min exercise impulse represented excess anabolic stimulation, and that each spaced 20-min impulse could stimulate osteogenesis, thereby doubling the anabolic response of a 40-min exercise bout. This hypothesis was supported by the results of previous study [19], where each of the two 40-min post-meal exercise bouts spaced by 7 h, produced an increase in CICP concentration, thus doubling the effect of a single 40-min bout. As our results showed (Figure 4) that a 20-min exercise impulse, either uphill or downhill, was insufficient to induce an anabolic response, and so our testing of the effects of restored mechanosensitivity failed. On the other hand, our 40-min downhill exercise bout produced its anabolic effect in the absence of exercise but simultaneously with, and at the time the second 20-min post-meal exercise bouts took place (Figure 4). The robust decline in CTX concentration after the mid-day meal at the time no exercise was taking place (Figure 3B, Figure 6A and Table 6) suggests restoration of sensitivity not only to mechanical stimulation but also to nutrient stimulation. This interpretation requires testing by spacing two 40-min downhill post-meal exercise bouts at 08:00 and 13:00 h to see whether the magnitude of CTX suppression is twice as large as in a single 40-min bout.

### 4.2. Testing of Hypothesis 2: The Importance of Pre-Exercise Meals

The central role of nutrient energy in support of bone growth is well established. There is ample evidence that dietary restriction and weight loss precipitate bone mineral loss in women athletes [40] with a 16.5 g of bone mineral loss for each kg of body-fat loss [41]. This general effect of fasting is seldom considered relevant in exercise studies that predominantly focus on mechanical loading. It is largely mediated by the failure of CTX concentration to decline during its mid-afternoon nadir [42,43]. Conversely, food ingestion suppresses bone resorption primarily through actions of a postprandial gut peptide GLP-2 [44,45]. GLP2 suppresses CTX concentration more than concentrations of OC, PINP, and BALP [46], and the ability of injected GLP-2 to reduce CTX concentration is dose-dependent [47]. While the involvement of nutrition in bone mass and mineral metabolism is generally accepted, our present and previous studies [19], along with the recent mouse study [20], are novel in that they demonstrate the close short-term temporal dependence between nutrient intake and enhanced effectiveness of exercise to stimulate osteogenesis.

Therefore, the essential requirement for the pre-exercise meal for amplifying the anabolic effect of exercise needs to be emphasized. A meal eaten one to two hours before exercise was necessary to elicit an anabolic response in the present and previous study [19], while the same exercise parameters in pre-meal exercise failed to do so [19]. The key difference between two 40-min downhill trials, both initiated 08:00 h, of which one suppressed afternoon CTX concentration (Figure 3B), and the other did not [19], is that in the former, the morning exercise was performed one hour after eating, and in the latter it took place before eating. We can only speculate whether fast walkers at level ground [18], who also initiated exercise at 08:00 h, ate a breakfast before exercising at the mall, as a way of explaining how their high speed and momentum and a 45-min impulse over four months resulted in suppression of bone mineral loss.

Our specific test of Hypothesis 2 was whether a pre-exercise meal would consistently produce an anabolic response in the morning after the 07:00 h meal as in the afternoon after the 13:00 h meal. In the present study, we observed no immediate post-meal increases in anabolic responses to the 07:00 h meal preceding the 08:00 h exercise in either spaced 20-minite trials or in the single 40-min trial (Figure 4, Figure 6B–D). Only exercise following the mid-day meal increased the anabolic response. Several studies provide three plausible explanations for this observation. The first one has to do with the relationship between the timing of meals and exercise in the present study and the timing of the CTX circadian rhythm. CTX concentration peaks between midnight and the early morning hours (between 24:00 and 05:00 h) and declines to its nadir in late afternoon between 14:00 and 18:00 h [48,49,50]. The 08:00 h exercise an hour after an 07:00 h breakfast, coincides with the high early morning circadian CTX concentrations (Figure 6A) which may be suppressing immediate anabolic effects of post-meal exercise. Second, in a study with remarkable similarities in the meal quality and timing to the present study, GLP-2 (and GIP) AUCs were only about 50–60% as high after 09:00 h breakfast as they were after the lunch [51]. This circadian difference in GLP-2 action may also have diminished the effectiveness of this peptide to suppress CTX after the morning meal. Finally, hunger and digestive physiology also appear to follow a circadian pattern. Hunger level, and possibly the digestive-absorptive processes, are lowest in the morning despite the long overnight fast. They reach peak between 13:00 and 19:00 h. This may potentially also contribute to the absence of a bone response to the morning post-meal exercise bouts [52].

Besides the evidence for a requirement for a pre-exercise meal as the reason for robust increases in CICP concentrations to 40-min exercise initiated at 11:00 h and 18:00 h in the previous study [19], two confounding factors cloud the conclusion. 

First, the health status of the subjects in the the previous and the present study, duration of pre-meal fast, and the circadian period of exercise, were all different. Fasting blood glucose in healthy postmenopausal women in the present study was a normal 94 mg/dl compared to elevated level of 112.7 ± 3.8 mg/dl in the diabetic subjects in the previous study. Hyperglycemia inhibits the production and release of bone formation markers PINP and osteocalcin [53]. In the 11:00 and 18:00 h post-meal trials, the HOMA-IR measures of insulin resistance were significantly reduced within one hour of both uphill and downhill exercise but remained high in exercise trials carried out before eating. It is therefore likely that a post-exercise decline in HOMA-IR reduced the inhibitory effect of hyperglycemia on markers of bone formation. Second, the fasting interval was 15 h long before the pre-exercise meals at 10:00 and 17:00 h in the previous study [19] while in the present study the pre-exercise fast duration was 11 h long as three daily meals were available. A recently published study with mice demonstrated that a 16-h fast before refeeding greatly amplified the anabolic effect of mechanical loading of a limb bone [20], suggesting that a longer pre-meal fast may have contributed to the anabolic outcome in the previous study.

### 4.3. Testing of Hypothesis 3: A Case for Circadian Interference in Bone Anabolism

Two results in the present and the previous study [19] suggest that circadian timing of exercise could affect its bone anabolic effects. First, CICP concentration increased when post-meal 40-min exercise was performed at 11:00 and 18:00 h either uphill or downhill but not when the same exercise was performed at 08:00 and 13:00 h [19]. While we attributed the absence of an anabolic response to exercise at 08:00 and 13:00 h in the previous study [19] to not eating a pre-exercise meal, we did not consider a possible circadian influence as the contributing cause. Additional research may be needed to determine whether exercising at 11:00 and 18:00 h selectively stimulates an increase in CICP rather than a suppression of CTX, does so in healthy as well as diabetic postmenopausal women, and why it delivers an osteogenic effect exercising twice separated by a 7-h interval either uphill or downhill. 

Second, unlike the 08:00 h exercise onset in the previous study which occurred before eating, in the present study the 08:00 h morning exercise occurred an hour after eating. Yet, unlike an equivalent CICP response to either uphill or downhill exercise performed one hour after the meals at 11:00 and 18:00, only downhill 40-min exercise in the present study suppressed CTX concentration. The effect suggested a circadian interference, and Hypothesis 3 was not supported. To analyze the potential circadian interference, it is useful to review three features of the PTH secretion and actions, as this hormone orchestrates both bone mineral loss and accretion.

The first remarkable feature of PTH action is the hormone’s contrasting capacity to either increase or decrease bone mineral content in response to body energy balance or calcium concentration in the blood. The previously mentioned effects of variable availability of nutrition (and calcium) on bone mineral mass are largely orchestrated by changes in PTH concentration. Second, PTH also has the capacity to switch between these opposing actions depending on different patterns of its release and concentration. Intermittent pulses of PTH promote bone mineral preservation and accrual, while sustained elevation of PTH leads to the bone mineral resorption [31,54,55]. The switching mechanism resides in the intermittent PTH stimulus which leads to preservation and increase in osteoblast numbers and lengthens the expression and action of a Runx2 gene. In the absence of this pulsatile secretory pattern, the sustained increase in its concentration leads to increased number and activity of bone-resorbing osteoclasts [54,56]. The third important feature of PTH action is its capacity to potentiate the bone anabolic effects of mechanical loading [57,58,59,60,61] or vibration [62] independently of its responsiveness to blood calcium concentrations.

The critical role of the timing of PTH pulses or rise in concentration on the circadian pattern of CTX rhythm in this phenomenon was discovered only relatively recently [63,64]. In two studies, osteoporotic postmenopausal women exhibiting low blood PTH concentrations were treated with the PTH analog teriparatide (TPTD) at two circadian times, 08:00 h or 20:00 h, for either one day [64] or 12 months [63]. Evening TPTD treatment for one day left the circadian CTX rise between 23:00 and 08:00 h unchanged from the control trial. However, morning 08:00 h TPTD treatment for one day abolished the nocturnal circadian CTX rise and acrophase. In the 12-month study, morning TPTD administration resulted in the larger increase in the lumbar spine BMD (9.1%) than the evening administration (4.8%). These findings help interpret our failure to observe any anabolic effect in the 40-min uphill exercise in the present study (Figure 3 and Figure 4). The post-exercise pattern of PTH concentrations differed between the 40-min downhill and uphill groups (Figure 7A). Although both groups displayed an exercise-associated peak one hour post exercise, elevation of post-exercise PTH concentration in the uphill trial was sustained and significantly higher than during downhill exercise (51.0 ± 2.5 vs. 43.4 ± 1.9 pg/mL, respectively, t_(df = 14)_ = 6.198, *p* < 0.0001). This difference was even greater in the previous study [19] with the mean post-exercise PTH concentration uphill being significantly higher than in the downhill trial (56.2 ± 1.2 vs. 34.6 ± 0.5 mg/dl, respectively, t_(df = 12)_ = 5.149, *p* = 0.0003). The combined results of the two TPTD studies, and our results on differences in post-exercise PTH concentrations in the present and the previous study, make it highly probable that the morning administration of either PTH or elicitation of PTH peak by downhill exercise suppressed the nocturnal circadian peak of CTX, while the sustained increase in post-exercise PTH concentration after 40 Up exercise failed to do so. It is therefore reasonable to conclude that the failure of the 40-min uphill exercise initiated at 08:00 h was a consequence of circadian interference through different PTH secretory patterns over the morning CTX secretory rhythm.

## 5. Study Limitations

This study has some limitations both in its methodology and design. Methodological limitations include, first, insufficient number of BALP measurements making this bone marker inadequately powered for a significant outcome (Figure 3D and Figure 4C). The statistical power was marginal in several other contrasts where no treatment, but only significant interactions were obtained, (Table 4 and Table 7). Second, the first insulin measurement at the start of 08:00 h exercise represented its first postprandial peak making insulin results less clear than they would have been if we used a fasting baseline. Third, glucose was not measured throughout the study reducing the ability to assess changes in insulin resistance. Finally, in both the present and previous study, we provided a 60% high-carbohydrate diet that has been recommended to U.S. consumers over the past 10 years by Department of Health and Human Services [65]. We demonstrated that this diet exacerbates insulin resistance when two bouts of 2 h moderate-intensity exercise are performed one hour before the 10-h and 17-h meals [66]. By simply reducing the carbohydrate content of the diet from 60% to 30%, the third exposure to the reduced-carbohydrate meal lowered plasma insulin and HOMA-IR measure of insulin resistance by approximately 30% in both the sedentary and exercise trials, and the effect was mediated by about 40% reduction of the insulinotropic gut peptide GIP. Thus, the macronutrient composition of the diet could have affected some of our results. 

Absence of two spaced 40-min exercise trials was the first design limitation which precluded our testing of Hypothesis 3. Second, this study limits our ability to generalize the results to men at risk of osteoporosis as we focused in this study on postmenopausal women. Third, matching of subjects in the five groups by the duration of time from the onset of menopause was not even as the sedentary and 40-min uphill groups were postmenopausal longer than the other three groups. This too could have affected our results as bone-mineral turnover rates differ during the first 10 years after the onset of menopause. 

## 6. Conclusions

Despite the stated limitations, this study provides novel recommendations for exercise protocols that can prevent or mitigate postmenopausal osteoporosis in sedentary non-athletic women. We provide exercise and dietary parameters for exercise on level ground or on a −6° downhill slope initiated at 08:00 h. They include a walking speed of between 6.1 and 6.7 km/h or 1.7 and 1.9 m/s, respectively, which produces a momentum of between 119 and 137 kg*m/s. Such exercise is effective in reducing concentrations of the resorptive bone marker CTX and actual bone resorption only if it is carried out for between 40 min in downhill circumstances or 45 min on level ground. It is possible that a greater anabolic effect could be achieved with two 40-min downhill exercise bouts performed at 08:00 and 15:00 h after two daily meals, suggested by apparent restoration of sensitivity to nutrients seven hours after the first post-meal exercise. The principal stimulus for the bone anabolic effect is exercise, but the prerequisite for a pre-exercise meal demonstrates the requirement for nutrient facilitation. Definition of a pre-meal dietary requirement and the above biomechanical parameters for exercise at 08:00 h may provide guidance in prevention of postmenopausal osteoporosis.

## Figures and Tables

**Figure 1 nutrients-13-03727-f001:**
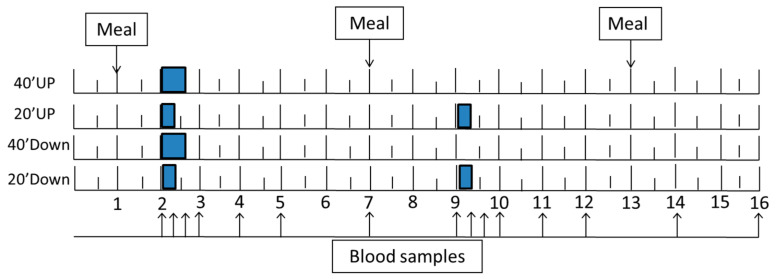
Experimental design. A single 40-min exercise bout, and the first of the two spaced 20-min exercise bouts, started at 08:00 h (point 2), while the second of the spaced bouts started 7 h later at 15:00 h (point 9). Of the three meals, the morning and mid-day meals were taken at 07:00 h (point 1) and 13:00 h (point 7), and the evening meal at 19:00 h (point 13). Arrows indicate times of blood collection.

**Figure 2 nutrients-13-03727-f002:**
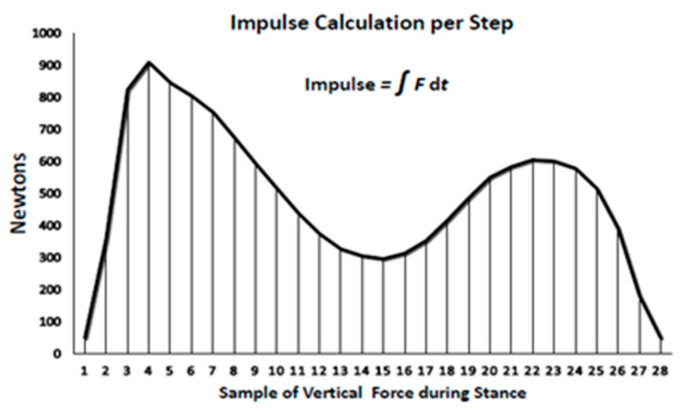
Impulse calculation per step.

**Figure 3 nutrients-13-03727-f003:**
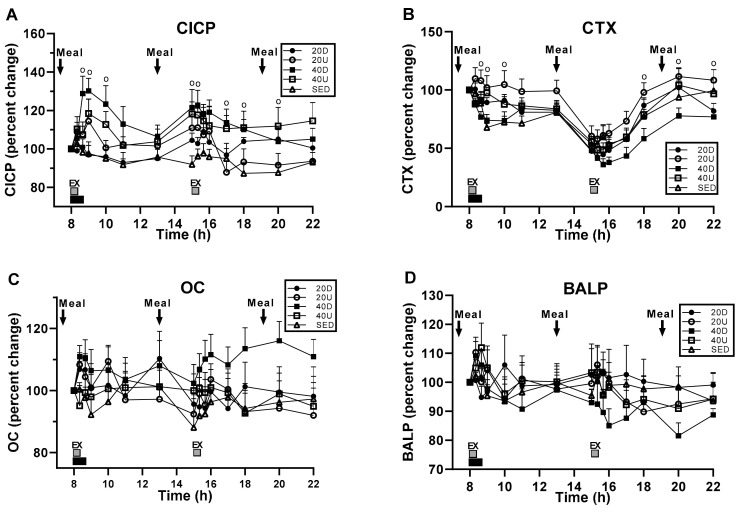
Time course of bone marker responses. CICP serum concentration (**A**) was significantly higher in 40 Down than in the SED group, and CTX concentration (**B**) significantly lower in 40 Down than in the 20 Up group. No changes in OC (**C**) and BALP (**D**). For CICP and CTX concentrations, “o” marks time slices of significant group differences.

**Figure 4 nutrients-13-03727-f004:**
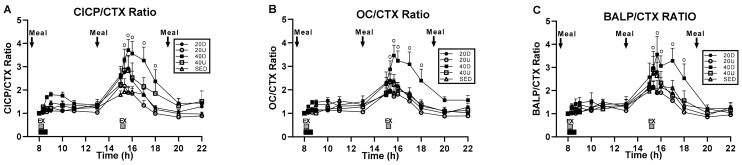
Changes in the ratios between the three markers of bone formation and the marker of bone resorption. CICP/CTX ratio (**A**) was significantly higher for 40 Down than for SED and 20 Up groups, and OC/CTX ratio (**B**) for 40 Down than for SED, 20 Up, and 40 Up groups. BALP/CTX ratio (**C**) were only significant difference between groups at time slices after the second exercise bout.“o” marks time slices of significant group differences.

**Figure 5 nutrients-13-03727-f005:**
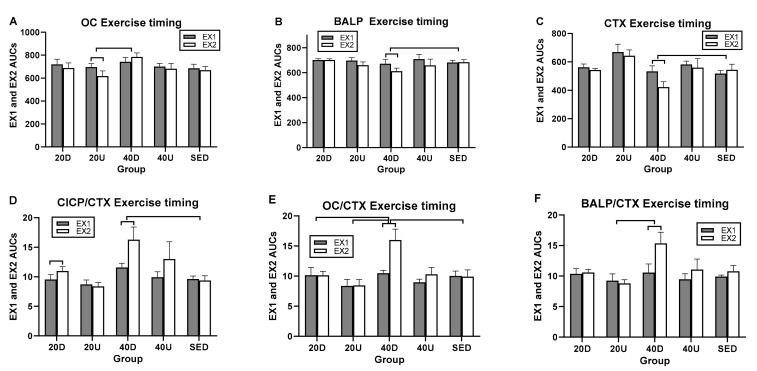
Between- and within-group comparisons for timing effects of EX1 and EX2 AUCs on markers of bone formation and their ratios. Between-group differences in EX AUCs were seen between 40Down trial and SED in BALP (**B**), CTX (**C**), CICP/CTX (**D**), and OC/CTX (**E**) trials; between 40 Down and 20 Up trials in OC (**A**), OC/CTX (**E**), and BALP/CTX (**F**); and 20 Down in OC/CTX (**E**). Within groups, EX1 CTX AUCs were higher than EX2 AUCs for OC (**A**), BALP (**B**), and CTX (**C**). EX2 AUCs were higher than EX1 AUCs for the three bone marker ratios (**D**–**F**). D = Down exercise AUC, U = Up exercise AUC, S = sedentary trial exercise AUC.

**Figure 6 nutrients-13-03727-f006:**
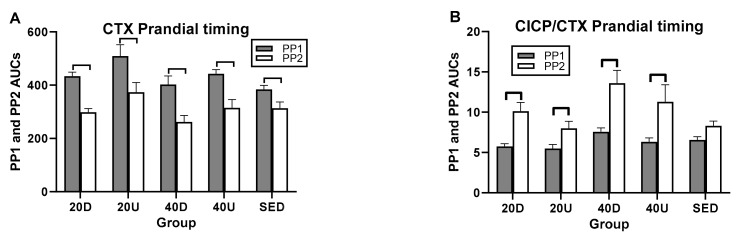
Comparisons between AUCs for two postprandial periods, PP1 and PP2 for CTX (**A**) and for the ratios of markers of bone formation over bone resorption (**B**–**D**). D=Down prandial AUC, U = Up prandial AUC, S = sedentary prandial AUC.

**Figure 7 nutrients-13-03727-f007:**
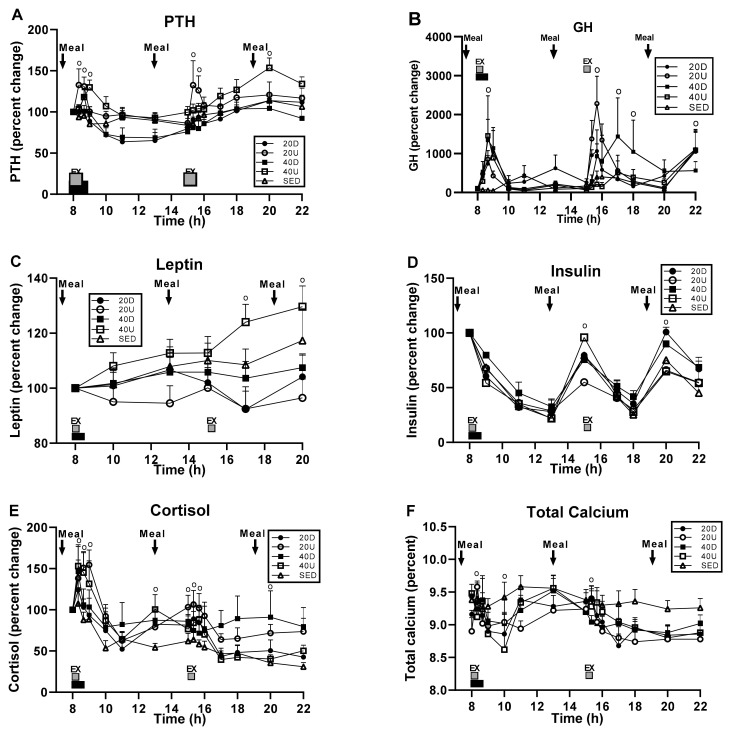
Effects of exercise and meals on the percent changes in hormone and in serum calcium concentrations. Treatment effect was significant only for PTH (**A**), time effects were significant for all five hormones (**A**–**E**), and the interaction between time and treatment for all but leptin (**A**,**B**,**D**–**F**). Group differences were significant at time slices marked with “o”. Total calcium concentrations in the four exercise groups were lower than in the SED group (panel **F**).

**Figure 8 nutrients-13-03727-f008:**
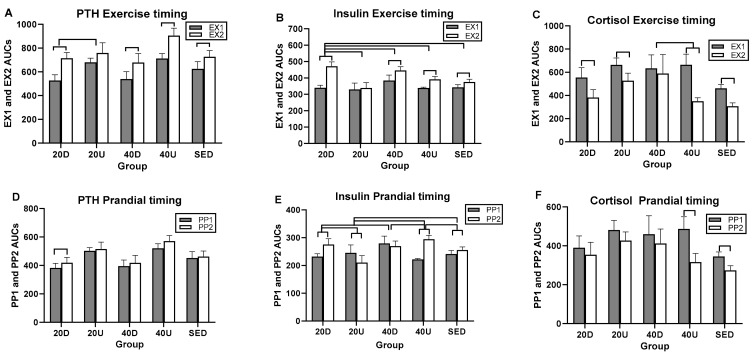
Between- and within-group comparisons for timing effects of EX1 and EX2 and PP1 and PP2 AUCs on PTH (**A**,**D**), insulin (**B**,**E**), and cortisol (**C**,**F**). D = down AUCs, U+Up AUCs, S = sedentary AUCs.

**Table 1 nutrients-13-03727-t001:** Composition of the three meals provided to study subjects.

**Breakfast 7:00 am, 25% of Calories for a 2000-Cal Daily Intake**					
Menu item	Amount	cals	pro	fat	carb
Cheerios	17 gm box	60	2	1	13
2% milk	8 oz carton	120	8	5	11
Cranberry muffin	One	164	4	6	27
Margarine	1 tub	36	0	4.1	0
Orange juice	two 4 oz boxes	120	0	0	28
total		500	14	16.1	79
**Lunch 1:00 pm, 35% of Calories**					
Menu item	Amount	cals	pro	fat	carb
Spaghetti cooked	100 gm	141	4.7	0.6	28
Ragu sauce	125 gm	80	2	3	10
Parmesan cheese	8 gm	34.4	3	2.3	0.3
Meat balls	40 gm	99	7	7	2.8
Green beans	91 gm	25	1	0	6
Tossed salad	45 gm	6	0.6	0	1
Sysco Italian dressing	12 gm	45	80	4.5	1
Wheat roll	30 gm	77	42.7	1.2	13.8
Margarine	1 tub	36	0	4.1	0
Sherbet	4 oz container	90	0	0	15
Sprite	255 gm	100	0	0	26
total		733.4	21	22.7	103.9
goal		700	26	19	105
**Dinner 7:00 pm, 40% of Calories**					
Menu item	Amount	cals	pro	fat	carb
Chicken breast	75 gm	118	22.5	2.7	0
Rice	120 gm	133	2.4	0	29
Herbed au jus	2 T	25	1	1	3
Carrots	86 gm	31	1	0	7
White roll	30 gm	71	1.9	1.4	12.8
Margarine	1 tub	36	0	4.1	0
Berry applesauce	1 container	90	0	0	23
Cheese Danish	One (2.25 oz)	231	3	15	21
Apple juice	180 gm	91	0	0	22.5
total		826	31.8	24.2	118.3
goal		800	30	22	120

Cals = calories, Pro = protein, Carb = carbohydrate.

**Table 2 nutrients-13-03727-t002:** Subject characteristics.

Variable	Sedentary	Uphill 20/20 min	Downhill 20/20 min	Uphill 40 min	Downhill 40 min	F_(df = 4,35)_; *p*
Subjects	*N* = 8	*N* = 8 (1 AA)	*N* = 8 (1 A)	*N* = 8	*N* = 8	
Age (years)	57.4 ± 1.7	58.4 ± 1.0	58.1 ± 1.5	59.9 ± 6.4	55.1 ± 1.0	1.82; 0.147
Menopause (y)	11.5 ± 2.7	6.6 ± 2.6	5.4 ± 1.6	12.8 ± 2.7	5.1 ± 1.1	2.453; 0.064
Weight (kg)	73.6 ± 3.8	74.6 ± 3.6	69.8 ± 2.9	67.8 ± 7.8	63.6 ± 5.3	0.8698; 0.4917
LBM (kg)	40.4 ± 1.8	42.1 ± 1.0	39.8 ± 1.3	38.8 ± 4.5	36.9 ± 3.1	0.0744; 0.9284
Body fat (%)	42.0 ± 2.3	40.0 ± 2.8	39.8 ± 2.0	39.4 ± 4.5	35.1 ± 3.0	0.2473; 0.9094
BMI (kg/m^2^)	26.2 ± 1.0	27.7 ± 1.0	27.1 ± 1.2	26.0 ± 2.9	23.9 ± 2.1	0.5956; 0.6682
BMD, body (g/cm^2^)	1.2 ± 0.0	1.1 ± 0.0	1.1 ± 0.0	1.1 ± 0.1	1.2 ± 0.0	0.7530; 0.5628
BMC, body (kg)	2.6 ± 0.1	2.7 ± 0.1	2.5 ± 0.1	2.4 ± 0.3	2.2 ± 0.2	0.620; 0.5434
Z score, body	0.7 ± 0.3	1.0 ± 0.3	0.5 ± 0.5	0.9 ± 0.2	1.0 ± 0.2	0.4921; 0.7415
BMD spine (g/cm^2^)	1.1 ± 0.1	1.1 ± 0.1	1.1 ± 0.1	0.9 ± 0.1	1.1 ± 0.0	1.034; 0.4037
BMC spine L1–L4 (g)	67.4 ± 7.1	58.1 ± 2.3	57.8 ± 1.4	51.7 ± 6.4	52.8 ± 5.3	0.8883; 0.4811
Z score spine	0.3 ± 0.6	0.2 ± 0.4	−0.2 ± 0.6	0.1 ± 0.2	0.1 ± 0.1	0.2038; 0.9346
BMD hip (g/cm^2^)	0.9 ± 0.0	0.9 ± 0.1	0.9 ± 0.0	0.9 ± 0.1	1.0 ± 0.0	2.024; 0.1124
BMC hip (g)	28.6 ± 1.1	31.4 ± 0.9	28.6 ± 1.1	28.2 ± 3.4	27.8 ± 2.4	1.225; 0.3053
Z score hip	−0.1 ± 0.2	0.1 ± 0.4	0.0 ± 0.4	0.3 ± 0.2	0.4 ± 0.3	0.8705; 0.4457
ṼO_2_ max (mL O_2_/min)	1847.2 ± 178.0	2087 ± 197.6	1793.0 ± 112.2	1646.4 ± 162.0	1891.9 ± 179.2	1.072; 0.3851
ṼO_2_ max (mL O_2_/(kg*min))	25.8 ± 1.8	28.2 ± 3.3	26.8 ± 2.1	23.7 ± 3.3	24.4 ± 2.4	0.6611; 0.6231
Basal serum PTH (ng/mL)	42.6 ± 1.5	44.6 ± 4.2	49.9 ± 6.0	39.5 ± 4.1	57.6 ± 7.6	1.557; 0.2073
Basal serum TSH (ng/mL)	2.3 ± 0.4	2.3 ± 0.3	3.1 ± 0.3	1.9 ± 0.2	2.8 ± 0.6	1.953; 0.1235

A = Asian subject; AA = African American subject; BMC=bone mineral content; BMD=bone mineral density; BMI = body mass index; LBM=lean body mass; PTH = parathyroid hormone; TSH = thyroid stimulating hormone; Z score = number of standard deviations below the bone mineral density normalized for gender and age.

**Table 3 nutrients-13-03727-t003:** Exercise outcome.

Variable	Uphill 20/20 min	Downhill 20/20 min	Uphill 40 min	Downhill 40 min	F_(df = 3,28)_; *p*
Subjects	*N* = 8	*N* = 8	*N* = 8	*N* = 8	
Treadmill slope (degrees)	7.9 ± 1.4	−6.0 ± 0.0	8.8 ± 1.4	−6.0 ± 0.0	48.32; <0.0001
Walking speed (kph)	4.4 ± 0.4	7.0 ± 0.4	4.9 ± 0.5	6.4 ± 0.1	20.25; <0.0001
Walking speed (m/s)	1.3 ± 0.0	1.9 ± 0.1	1.2 ± 0.2	1.8 ± 0.0	20.28; <0.0001
Distance walked (km)	2.9 ± 0.3	4.6 ± 0.3	3.2 ± 0.0	4.3 ± 0.1	20.29; <0.0001
Relative effort (%ṼO_2_ max)	76.9 ± 2.5	46.5 ± 2.9	67.3 ± 5.9	46.7 ± 3.2	27.7; <0.0001
Heart rate (bpm)	135.6 ± 2.2	118.4 ± 3.3	121.7 ± 10.5	119.3 ± 3.7	10.33; <0.0001
RPE	12.2 ± 1.4	12.0 ± 1.0	11.2 ± 1.0	11.4 ± 0.7	0.1739; 0.9138
Steps in 40 min	4103.7 ± 350.9	5196.9 ± 70.1	4568.1 ± 82.4	5209 ± 100.3	20.38; <0.0001
Peak pressure (KPa)	265.6 ± 17.3	364.2 ± 20.7	242.6 ± 20.8	296.0 ± 11.0	8.282; 0.0004
Relative peak pressure (KPa/kg)	3.6 ± 0.3	5.3 ± 0.4	4.0 ± 0.4	4.3 ± 0.4	5.813; 0.0032
Peak GRF (N)	839.9 ± 30.8	1017.1 ± 48.7	780.1 ± 44.0	1109.5 ± 46.2	14.36; <0.0001
Relative GRF (N/kg)	1.1 ± 0.1	1.6 ± 0.1	1.3 ± 0.1	1.4 ± 0.1	7.417; 0.0008
Momentum ((kg*m)/s)	100.1 ± 4.6	134.6 ± 9.0	92.9 ± 4.1	127.4 ± 4.6	10.94; <0.0001
Impulse or force–time integral (N*s)	1,007,880 ± 36,960	1,220,520 ± 58,440	1,872,240 ± 105,600	2,662,800 ± 110,880	33.0; *p* < 0.0001

HR = heart rate; GRF = ground reaction force; KPa = kilopascal; N = Newton; RPE = ratings of perceived exertion.

**Table 4 nutrients-13-03727-t004:** Group differences resulting from timing of meals and exercise for the three markers of bone formation, the marker of bone resorption, and their ratios.

Markers or Their Ratio	Treatment	Time	Interaction	Slices (Times) Significant at >0.05
CICP	F_(df = 4,35)_ = 3.57, *p* = 0.00153	F_(df = 14,490)_ = 3.97, *p* < 0.0001	F_(df = 56,490)_ = 1.37, *p* = 0.046	8:40, 9:00, 10:00; 15:00, 15:20, 17:00, 18:00, and 20:00 h.
OC	NS	NS	NS	
BALP	NS	NS	NS	
CTX	F_(df = 4,35)_ = 4.81, *p* = 0.0034	F_(df = 14,490)_ = 48.51, *p* < 0.0001	NS	8:40, 9:00, 10:00 h;18:00, 20:00, 22:00 h
CICP/CTX	F_(df = 4,35)_ = 3.74, *p* = 0.0123	F_(df = 14,490)_ = 3.74, *p* = 0.0123	F_(df = 56,490)_ = 1.83, *p* = 0.0004	15:20, 15:40, 16:00, 17:00, 18:00 h
OC/CTX	F_(df = 4,35)_ = 4.39, *p* = 0.0056	F_(df = 14,490)_ = 38.65, *p* < 0.0001	F_(df = 56,490)_ = 2.12, *p* < 0.0001	15:20, 15:40, 16:00, 17:00, 18:00 h
BALP/CTX	NS	F_(df = 14,280)_ = 31.75, *p* < 0.0001	F_(df = 56,280)_ = 1.63, *p* = 0.0057	15:20, 15:40, 16:00, 17:00, 18:00 h

**Table 5 nutrients-13-03727-t005:** Between group and within-group timing differences in exercise AUCs.

Markers/Ratios	Overall Timing Effect	Between-Group Difference (95% CIs)	Within Group Difference t_(df = 7)_ *l(*p*)
CICP	F_(df = 4)_ = 3.16, *p* = 0.0257	20 D > 20 U (0.0308, 0.444)	NS
OC	F_(df = 4)_ = 3.30, *p* = 0.0213	40 D > 20 U (0.038, 0.341)	EX1 > EX2 for 20 U (2.98–0.02)
BALP	F_(df = 4)_ = 3.19, *p* = 0.0353	40 D > S (0.187, 0.0004)	EX1 > EX2 for 40 D (5.40–0.006)
CTX	F_(df = 4)_ = 2.44, *p* = 0.0653	40 D > S (0.5953, 0.0293)	EX1 > EX2 for 40 D (2.46–0.04)
CICP/CTX	NS	40 D > S (0.0045, 0.6565)	EX2 > EX1 for 40 D (2.70–0.03)
OC/CTX	NS	40 D > S (1223, 0.7126) 40 D > 20 D (0.058, 0.648) 40 D > 20 U (0.07, 0.661)	EX2 > EX1 for 40 D (4.1–0.005)
BALP/CTX	NS	40 D > 20 U(0.0127, 0.78)	EX2 > EX1 for 40 D (3.68–0.021)

* For BALP and BALP/CTX, t_(df = 4)._ D = Down EX AUC, U = Up EX AUC, S = SED EX AUC.

**Table 6 nutrients-13-03727-t006:** Between group and within-group timing differences in postprandial AUCs.

Markers/Ratios	Overall Timing Effect	Between-Group Difference (95% CIs)	Within Group Difference t_(df = 7)_ * *p*
CTX	NS	NS	PP1 > PP2 for all (t_(df = 7)_ 4.26 to 8.53– *p* = 0.004 to 0.0001)
CICP/CTX	NS	NS	PP2 > PP1 # PP2 > PP1 # PP2 > PP1 #
OC/CTX	NS	40D > S (0.021,0.75)	
BALP/CTX	NS	NS	

* t_(df = 4)_ for BALP/CTX; # differences for exercise groups only; D = Down exercise AUC; U = Up exercise AUC, S = sedentary trial exercise AUC.

**Table 7 nutrients-13-03727-t007:** Group differences for five hormones resulting from timing of meals and exercise.

Hormones	Treatment	Time	Interaction	Slices (Times) Significant at >0.05
PTH	F_(df = 4,35)_ = 3.13, *p* = 0.0267	F_(df = 14,490)_ = 19.02, *p* < 0.0001	F_(df = 56,490)_ = 2.12, *p* < 0.0001	09:00, 10:00; 11:00; 13:00; 17:00; 20:00 h.
GH	NS	F_(df = 14,488)_ = 13.07, *p* < 0.0001	F_(df = 56,490)_ = 3.10 *p* < 0.0001	08:20; 08:40; 9:00; 15:20; 15:40; 16:00 h.
Leptin	NS	F_(df = 14,490)_ = 3.65, *p* = 0.0036	NS	17:00; 20:00 h
Insulin	NS	F_(df = 14,490)_ =162.5, *p* < 0.0001	F_(df = 56,490)_ = 3.17, *p* < 0.0001	09:00; 15:00 h; 20:00; 22:00 h
Cortisol	NS	F_(df = 14,490)_ = 29.13, *p* < 0.0001	F_(df = 56,490)_ = 1.50, *p* = 0.0137	08:20; 08:40; 9:00; 17:00; 20:00; 22:00 h.
Calcium, total (percent)	NS	F_(df = 14,490)_ = 14.10, *p* < 0.0001	F_(df =56,490)_ = 2.06, *p* < 0.0001	08:20, 10:00; 15:20 h.

**Table 8 nutrients-13-03727-t008:** Between group and within-group timing differences in exercise and postprandial AUCs.

Hormones	Overall Exercise Timing Effect	Between-Group Difference (95% CIs)	Within-Group Difference t_(df = 7)_-*p*
PTH	NS	20D−40U (0.00113–0.485)	EX2 > EX1 for 20D (4.68–0.0023) EX2 > EX1 for 40D (4.77–0.002) EX2 > EX1 for 40U (5.16–0.013) EX2 > EX1 for S (3.52–0.0098)
Insulin	F_(df = 4)_ = 9.67, *p* = 0.0001	20D−40D (0.0194,0.30431) 20D−40U (0.0436, 0.329) 20D-S (0.0917,0.3772) 20D-U (0.1472,0.4327)	EX2 > EX1 for 20D (13.58–0.0001) EX2 > EX1 for 40D (4.35–0.0034) EX2 > EX1 for 40U (3.08–0.0179) EX2 > EX1 for S (4.17–0.0042)
Cortisol	NS	40D−40U (0.0128–0.894)	EX1 > EX2 for 20D (6.16–0.0005) EX1 > EX2 for 20U (4.14–0.0043) EX1 > EX2 for 40U (7.48–0.0001) EX1 > EX2 for S (4.09–0.0046)
	**Overall Postprandial Timing Effect**		
PTH	NS	NS	PP2 > PP1 for 20D (4.61–0.0025)
Insulin	F_(df = 4)_ = 21.77, *p* = 0.0001	20U−40U (0.2904,0.5581) 20U−20D (0.172,0.4625 20U-S (0.7685.0.3674 20D−40D (003452,0.3251 40U−40D (0.153,0.44352) 40U-S (0.0683,0.35884)	PP2 > PP1 1 for 20D (4.27–0.0037) PP2 > PP1 for 20U (5.64–0.0008) PP2 > PP1 for 40U (5.44–0.0001) PP2 > PP1 for S (3.08–0.0178) F_(df = 7)_ = 3.08, *p* = 0.0179 F_(df = 7)_ = 4.17, *p* = 0.0042
Cortisol	NS	NS	PP1 > PP2 for 40U (5.44–0.001) PP1 > PP2 for S (3.12–0.0169)

D = downhill; U = uphill; S = sedentary.

## Data Availability

Data are available from the first author.

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
