# Peer review of "Anabolic Bone Stimulus Requires a Pre-Exercise Meal and 45-Minute Walking Impulse of Suprathreshold Speed-Enhanced Momentum to Prevent or Mitigate Postmenopausal Osteoporosis within Circadian Constraints"

_nutrients, 2021, doi:10.3390/nu13113727_

Round 1
Reviewer 1 Report
It is an interesting paper, but it seems like a part of wider study. It could be, however, a valuable piece of work, with clear aim, duration, results concerned on this study(I do not understand the value of 45 min in recommendations?), with the discussion concerned, in higher extent, on the other scientific papers.
Methodology also needs improvement. What what the reason for only 3 meals for the participants of the study?
What about snacking, what it permitted, controlled?
What about the diet composition before and during experiment, concerned on components important for healthy bones?
Additionally, were the meals the same for the women with different BMI?
What about the biochemical parameters values before the experiment - are they important, taking into consideration for example the women age as well as time from menopause?
Moreover, what about the results in long period of time? What period of time is needed to ensure the beneficial effects for individual woman at this age period? What about dietary recommendations?
The Authors should consider these questions and add the text, where appropriate.
Author Response
We thank both reviewers for reading and critically evaluating our revised manuscript. Both had some good things to say about our manuscript, Reviewer 1 stating that ours was an interesting paper which could be a valuable piece of work, and Reviewer 2 stating that our data are interesting. We appreciate these evaluations as we are aware that our paper was out of the ordinary with an overload of biomechanical measurements and unusual concepts, but also with an important message that nutrition is essential in allowing the expression of the anabolic effects of exercise. Also that duration of an exercise bout, along with increased walking speed, are key ingredients for prevention of osteoporosis in postmenopausal women.
We would first like to comment on our general editorial actions. We received the MSWord manuscrIpt with very narrow text width and a mixture of Times New Roman (our original submission font) and Palatino linotype (Nutrients font) in both the text and tables. We took the liberty to extend the text width to full page, and correct the font to Palatino linotype wherever we noticed the mismatch. This worked well in the “track changes” (TC) version of the manuscript where the line numbers were preserved, but not in the clean copy (with “TC” accepted) where the line numbers were lost. Sorry about that.
Another general change we instituted was to delete reference to “speed-enhanced load” both in the title and in the text. This decision was based on our not finding strong support for the magnitude of ground reaction forces (as tested in uphill and downhill walking) having a significant effect on osteogenesis. Our data suggest that it is the momentum, the force dependent on speed of movement that is the necessary requirement for osteogenesis. By deleting the term from both the title and the text, we streamlined and simplified the issues.
The following are our point-by point responses to reviewers’ recommendations for corrections and questions.
REVIEWER 1. Checked the following areas to be improved: Background and references in the introduction, Research design, Adequate description of methods, and conclusions.
- Reviewer did not understand the value of recommendation for the necessary 45-minutes of walking.
RESPONSE: The explanation can be found in lines 27-30 of the TC abstract and lines 423, 444-460 of the discussion. Our data unambiguously showed that only walking for 45 minutes on level ground at supra-threshold speed greater than 6 km/h (and for 40 minutes on a downhill treadmill) was it possible for post-meal exercise to induce bone anabolism. Cutting the impulse (duration of walking) by half) had no anabolic effect (Figure 4).
- Methodology concern was for us to explain the reason for only 3 meals in the study with no snacking permitted.
RESPONSE: In clinical studies (an in short experimental studies in general), it is necessary to establish a protocol for all variables to which all participants need to adhere in order to eliminate extraneous noise and variability. We selected three weight-maintenance meals for each subject based on their weight and provided them with meals of uniform macronutrient composition. Therefore allowing the subjects to have snacks would have increased their daily intake above their weight maintenance, and introduced experimental noise which we wanted to avoid.
- Additional methodology question: Were the meals the same with women with different BMI?
RESPONSE: Yes. BMI effects were part of between-subject variability. Standardized meals for subjects of all BMIs reduced this variability.
- Additional methodology question: What about biochemical parameters values before the experiment-are they important, taking into consideration for example the women age as well as time from menopause?
RESPONSE: We recruited ostensibly healthy postmenopausal women. Therefore our main concerns were (1) that they were not hypothyroid which is common in this age group. This would affect their energy expenditure. We therefore checked out their TSH, an indicator of thyroid hormone concentration. Additional concerns were their (2) PTH and (3) calcium concentrations to check against secondary hyperparathyroidism. This condition occurs when women have very low calcium intake or poor exposure to sun and vitamin D levels. This would potentially lead to sustained high PTH blood concentrations and baseline increase in bone resorption. That is why we measured PTH at the outset (see table 2 for values in all study groups) and found no group differences. While there are many other biochemical measurements one could check related to subjects’ use of medications, we did not try to measure them since the subjects were generally healthy. We were concerned about our failure to more adequately match women for the duration of time since menopause. As table 1 shows, sedentary and uphill 40-minute group were menopausal for a longer period than the other three groups (p value close to significance). However, there were no group differences in bone mineral parameters we measured with DXA (Table 2), suggesting that the length of time since menopause did not adversely affect the results.
- Conclusion question: What about the results in long period of time? What period of time is needed to ensure the beneficial effects for individual woman at this age period? What about dietary recommendations?
RESPONSE: This is a relevant question, but the answer requires a different type of study such as a randomized controlled trial (RCT). In a RCT, the relevance of a certain type of diet or of dietary recommendation is compared to groups that do not get exposed to these variable. Our study was looking only at short-term effects of exercise and nutrition to provide postmenopausl women with recommendations on how to conduct their lives to prevent osteoporosis.

Reviewer 2 Report
The authors performed an interventional study whose primary aim was to assess how different exercise protocols can be effective in preventing postmenopausal osteoporosis. Data are interesting but there are comments from this reviewer as far as the methodology used to address the primary endpoint and data presentation.
Introduction
-This section should be shorten. For example, page 2: I suggest to move the reports on previous studies from the group in the discussion.
Methods
-Page 6: please detail the DXA machine used in the protocol. Also, as BMD was measured only at baseline, this should be specified here.
-Was baseline vitamin D status assessed in these patients? This could have effect particularly on serum PTH.
-Why cortisol, leptin, insulin and GH were measured? Do the authors have any data on serum sclerostin, a main mediator of bone anabolism?
Results
-Page 10: it is not clear whether the results reported in this page as far as bone markers are in response to exercise or meal or both. How it could be differentiated between the effects on bone markers of exercise intervention and those of meal?
Discussion
-This section should be extensively shorten.
Author Response
Authors response to reviews of manuscript Nutrients-1315391. October 2, 2021
We thank both reviewers for reading and critically evaluating our revised manuscript. Both had some good things to say about our manuscript, Reviewer 1 stating that ours was an interesting paper which could be a valuable piece of work, and Reviewer 2 stating that our data are interesting. We appreciate these evaluations as we are aware that our paper was out of the ordinary with an overload of biomechanical measurements and unusual concepts, but also with an important message that nutrition is essential in allowing the expression of the anabolic effects of exercise. Also that duration of an exercise bout, along with increased walking speed, are key ingredients for prevention of osteoporosis in postmenopausal women.
We would first like to comment on our general editorial actions. We received the MSWord manuscrIpt with very narrow text width and a mixture of Times New Roman (our original submission font) and Palatino linotype (Nutrients font) in both the text and tables. We took the liberty to extend the text width to full page, and correct the font to Palatino linotype wherever we noticed the mismatch. This worked well in the “track changes” (TC) version of the manuscript where the line numbers were preserved, but not in the clean copy (with “TC” accepted) where the line numbers were lost. Sorry about that.
Another general change we instituted was to delete reference to “speed-enhanced load” both in the title and in the text. This decision was based on our not finding strong support for the magnitude of ground reaction forces (as tested in uphill and downhill walking) having a significant effect on osteogenesis. Our data suggest that it is the momentum, the force dependent on speed of movement that is the necessary requirement for osteogenesis. By deleting the term from both the title and the text, we streamlined and simplified the issues.
The following are our point-by point responses to reviewers’ recommendations for corrections and questions.
REVIEWER 2. Checked the following areas to be improved: Background and references in the introduction and clarity of result presentation.
- On page 2: I suggest to move the reports from previous studies from the group in the discussion.
RESPONSE: We are sensitive to the criticism that we are bolstering the results of the present study with results from 2 previous studies. This issue was the reason for an earlier request from Nutrients chief editor that we substantially revise our original manuscript (which we did) and remove these references. In response to the present recommendation, we removed all factual details about the two previous studies from the introduction (see deleted lines 64 -70 in TC version). We now only mentioned that one study demonstrated the requirement for increased walking speed, and the other the requirement for pre-exercise meal. More detailed explanation of the two studies was now moved to Discussion (lines 379-393). While we appreciate the general expectation that a research study should only provide details and results from that study, in our case, our reasoning and conclusions from the present study were critically complemented by the results and conclusions from the two previous studies. The full understanding of the results of the present study depends on a discussion of all three studies. This issue is now better situated in the discussion section.
- Methods request: please detail the DXA machine used in the protocol (page 6). Also was BMD was measured only at baseline, this should be specified here.
RESPONSE: We have originally provided DXA apparatus specifications in 2.2 General experimental Protocol (line 121-122 TC version) but did not repeat it in section 2.7 DXA measurements. In response to reviewer’s recommendation, we have now moved this information to section 2.7 (lines 176-177). Since ours was a one-day study, a single DXA measurement was performed to check bone mineral parameters in experimental groups (Table 2). We did not expect any changes in bone mineral parameters in a 16-hour study, so we did not make any additional comments about a single DXA measurement.
*Additional Methods request: Was baseline vitamin D status assessed in these patients? This could have effect particularly on PTH.
RESPONSE: We were aware about the possible complication of variable vitamin D, calcium, and PTH status. That was the reason (1) for our measuring PTH concentration at the baseline which did not differ between the groups (Table 2), and for our providing the subjects a 1g supplement of calcium and 600 IUs of vitamin D supplement in one oz of orange juice at 07:00 h at the start of the study (see Study design, lines 135-136). We assumed that a supplement of calcium and vitamin D would attenuate for 16 hours any possible variations in these two variables in our subjects.
*Additional Methods request: Why leptin, insulin, cortisol and GH were measured?
RESPONSE: See lines 104-106. All four of these hormones respond to exercise (25, 26, 27, 28, and 29, respectively). The first four support bone formation, and cortisol can block it (30, 31, 32, 33, and 34, respectively). Because were interested in their potential contribution to our exercise results, we measured their concentrations. Although none of these hormones appeared to change in a way that would suggest their influence on the exercise outcome, our measuring of PTH concentrations during and after exercise, provided a significant clue about the circadian interference in the anabolic effects of exercise.
*Additional Methods request: Do the authors have any data on serum sclerostin, a main mediator of bone anabolism?
RESPONSE: No. When we were designing our study, we mainly wanted to examine the evidence that our experimental paradigm affected bone anabolic responses. So we measured markers of bone formation and CTX, the marker of bone resorption. Now that we have provided information on the required biomechanical and nutritional parameters that are effective in suppressing CTX, measurement of sclerostin would be interesting and warranted.
*Clarity of result presentation: It is not clear whether the results reported in this page (page 10) as far as bone markers are in response to exercise or meal or both. How it could be differentiated between the effects of exercise intervention and those of meal?
RESPONSE: This is a great question that goes to the heart of the results of this study. It appears clear to us that the key variables that suppresses CTX in our study are components of the exercise paradigm, higher walking speed above 6 km/h and an impulse of between 40 and 45 minutes. Without these, no CTX suppression takes place (Figure 4). However, it is equally clear that without a pre-exercise meal, none of this exercise works (results of the [19] study and the very impressive results of the mechanical bone loading in mice being dramatically increased when the animals were refed after a 16-h fast ([20]). Therefore our conclusion is that the bone marker response is caused by specific parameters of exercise, but that the timing of nutritional support is essential.I have made statements to that effect both in the abstract . I added “The principal stimulus for the anabolic effect is exercise, but the prerequisite for a pre-exercise meal demonstrate the requirement for nutrient facilitation” (lines 33-34). And I added the same statement in Conclusions (lines 694-695).
As a post-script, we want to point out that in the present revision we substantially clarified, modified, and improved our three hypotheses, and thereby modified and improved their discussion. This may not have produced a great reduction in the length of the text, but we feel confident that it improved the clarity of result presentation.
